# Lactoferrin, Osteopontin and Lactoferrin–Osteopontin Complex: A Critical Look on Their Role in Perinatal Period and Cardiometabolic Disorders

**DOI:** 10.3390/nu15061394

**Published:** 2023-03-14

**Authors:** Emile Levy, Valérie Marcil, Sarah Tagharist Ép Baumel, Noam Dahan, Edgard Delvin, Schohraya Spahis

**Affiliations:** 1Research Centre, CHU Sainte-Justine, 3175 Sainte-Catherine Road, Montreal, QC H3T 1C5, Canada; 2Department of Nutrition, Université de Montreal, C. P. 6205, succursale Centre-ville, Montreal, QC H3C 3T5, Canada; 3Biochemistry &Molecular Medicine, Faculty of Medicine, Université de Montreal, C. P. 6205, succursale Centre-ville, Montreal, QC H3C 3T5, Canada

**Keywords:** milk-derived proteins, microbiota, cardiometabolism

## Abstract

Milk-derived bioactive proteins have increasingly gained attention and consideration throughout the world due to their high-quality amino acids and multiple health-promoting attributes. Apparently, being at the forefront of functional foods, these bioactive proteins are also suggested as potential alternatives for the management of various complex diseases. In this review, we will focus on lactoferrin (LF) and osteopontin (OPN), two multifunctional dairy proteins, as well as to their naturally occurring bioactive LF–OPN complex. While describing their wide variety of physiological, biochemical, and nutritional functionalities, we will emphasize their specific roles in the perinatal period. Afterwards, we will evaluate their ability to control oxidative stress, inflammation, gut mucosal barrier, and intestinal microbiota in link with cardiometabolic disorders (CMD) (obesity, insulin resistance, dyslipidemia, and hypertension) and associated complications (diabetes and atherosclerosis). This review will not only attempt to highlight the mechanisms of action, but it will critically discuss the potential therapeutic applications of the underlined bioactive proteins in CMD.

## 1. Introduction

The scientific and medical community is increasingly realizing the benefits of the proteins present in both the human breast and bovine milk. Besides being a crucial source of nitrogen and amino acids, this milk macronutrient also consists of bioactive peptides/proteins that are progressively recognized to perform remarkable functions [1,2]. Recently, the sophisticated development of a milk proteomics technique revealed 1606 proteins [3], of which a proportion is well established in the vast literature for its association with the development of the gastrointestinal (GI) tract, and the immune and central nervous systems. Despite this remarkable progress, efforts are still required to identify the other protein components of milk as the expression of many more than 10,000 genes was detected in mammary glands during lactation [4,5].

To date, few milk protein components have been largely probed in early life, but work is still needed to define their role in health and diseases later in life. Hence, the present topic is timely and worthy of exploration in complex diseases that take a heavy toll on the world’s population.

The main objective of this review is to take a critical look at two milk proteins: lactoferrin (LF) and osteopontin (OPN), which may influence the development of obesity and related disorders [6], known to have reached epidemic proportions worldwide, increase morbidity and mortality, and cause a significant socioeconomic burden. Since these two proteins show a high affinity for each other and form a bioactive (LF–OPN) complex with interesting properties, we will also focus on its potential functions as suggested by several groups.

First, we will document the spectrum of general properties and functions of each of the specific milk-derived bioactive components, particularly during the perinatal period. Second, we will provide a thorough overview of their implication in relevant intestinal mechanisms (i.e., intestinal barrier, permeability, microbiota), which regulate and impact the host’s cardiometabolic health. Third, we will focus on their influence on CMD, including obesity, metabolic syndrome (MetS), type 2 diabetes, and cardiovascular diseases (CVD), with a particular emphasis on clinical studies.

## 2. General Lactoferrin Properties

LF is an iron-binding monomeric glycoprotein of the transferrin family, which is substantially present in human and bovine milk [7], but also in lower amounts in saliva, tears, semen, and vaginal fluids [8]. Bovine and human LFs have features in common, including sharing the transferrin family, possessing identical iron-binding sites, and having two symmetrical homologous globular lobes (N-terminal lobe and C-terminal lobe). However, differences are also apparent, such as molecular weights (~80 and ~76 kDa, respectively), amino acid residues (689 and 691, respectively), and milk concentrations (0.02–0.35 and 1.0–3.2 mg/mL). In addition to binding nonheme iron, LF displays multiple biological activities and functions [9]. This multifunctional protein is highly resistant to GI digestion; promotes gut development; efficiently binds to specific membrane receptors of various tissues, including monocytes, lymphocytes, adipocytes, hepatocytes, enterocytes, and endothelial cells; and has the potential to regulate several intracellular signaling pathways following its intact absorption to the bloodstream [9,10]. LF is described as a pleiotropic protein with antioxidants, anti-inflammatory, antibacterial, antiviral, antifungal, anticancer, and immunomodulatory actions [9]. Among the several mechanisms for its beneficial impacts, intestinal microbiota seems largely involved with the potential to modulate CMD (Figure 1).

### 2.1. Lactoferrin and Intestinal Barrier

Early studies have shown that LF has the ability to bind to the epithelial layer of the GI mucosa of young children [41]. Other investigators reported the capacity of LF to protect the integrity of the gut barrier by linking to enterocyte and brush-border membranes, thus preventing the leakage and translocation of intestinal microbes to the bloodstream [25]. Additionally, LF may preserve the structural and functional epithelial layer by exerting protection against pathogens and toxic bacterial endotoxins that generally damage the tight junctions [42,43]. Importantly, LF molecules can form oligomeric complexes, which are instrumental for the maintenance of intestinal integrity, and influential in the protection against exogenous threats [44]. In fact, LF concomitantly stimulated intestinal epithelium growth and up-regulated the expression of jejunal tight-junction proteins such as zonula occludens and claudins in response to caloric restriction-mediated malnutrition [45]. The restoration of intestinal integrity, reflected by raised occludin mass and tight junction structure normalization, was caused by the inactivation of the MAPK pathway, in particular via the decrease of p38 and ERK1/2 phosphorylation [27]. The strengthening of the intestinal barrier was also accompanied by elevated alkaline phosphatase activity and transepithelial electrical resistance, the latter as an indicator of the strength of tight-junction proteins [46]. Confirmation has recently been obtained by Gao et al. [47], who reported LF protective effects not only on a tight junction through MAPK signaling, but also on endocytosis, adherens, and gap junctions, thereby assuring barrier structure and function. Moreover, using cross-omics analysis of transcriptome and proteome, the same authors unraveled the association of LF with an insulin receptor, cytoplasmic FMR1 interacting protein 2, dedicator of cytokinesis and ribonucleotide reductase regulatory subunit M2 proteins, which together contributed to the biological benefits of LF in the defense of the intestinal epithelium [47].

### 2.2. Lactoferrin and Intestinal Permeability

Generally, intact tight junction proteins impede the paracellular passage of molecules larger than ~600 kDa [48]. If LF undeniably preserves the intestinal barrier as reported above, it is expected that gut permeability would optimally be controlled as tight junctions constitute an integral paracellular seal [49]. The LF uptake resulted with lower cell permeability via the upregulation of myosin light-chain kinase expression in IEC-18 cells [50]. In other intestinal cell lines HT-29/B6 and T84, preincubation with LF ameliorated transepithelial resistance and attenuated cytokine-induced permeability, producing leaky gut [51]. In a similar way, LF was effective in reducing FITC–dextran permeability in response to bacterial-derived lipopolysaccharide (LPS) [52]. LF and its peptide fragments were also tested in mice with LPS-induced disruption of the intestinal tight junction structure and function [53]. Their intraperitoneal administration lessened permeability, inhibited colonic infiltration with polymorphonuclear leukocyte activity in colon tissue, and restored intestinal barrier integrity [53]. In a rat model of short bowel syndrome, bovine enteral LF supplementation increased villus height, crypt depth, and intestinal epithelial cell proliferation index, while preventing tight junction disruption and epithelium hyperpermeability [54]. Interestingly, engineered recombinant *Lactobacillus reuteri* expressing bovine LF peptide was able to reinforce the integrity of piglet’s gut barrier function as reflected by the elevation of zonula occludens and claudin proteins resulting from limited TLR4, Myd88, and myosin light-chain kinase protein expression [55]. Since inflammatory bowel diseases (IBD) are characterized by intestinal barrier disruption in response to epithelium injury and hyperpermeability, which augments host susceptibility to infection by pathogens [56,57], it seemed reasonable to determine whether LF (empowered with various biological functions) might restore gut homeostasis. This hypothesis was tested in murine IBD models (dextran sodium sulfate colitis model and TNF^ΔARE/+^ model of ileitis), where LF recombinant administration provided a protection against mucosal injury, largely documented by the reduction of crypt architecture abrasion, goblet cell deficiency, immune cell infiltration, inflammation, and hyperpermeability [58]. Another group assessed the impact of direct delivery of LF using *L. lactis* AMJ1543 on mice with dextran sulfate sodium-induced experimental colitis. Here again, LF administration clearly improved DSS-induced colon damage, which was substantiated by tight-junction protein overexpression and intestinal permeability alleviation [59].

In human, oral supplementation of 5 g LF and 50 mg indomethacin reduced the NSAID-mediated increase in small intestinal permeability, thereby providing a nutritional therapeutic tool in the treatment of hyperpermeability-associated disorders [60].

### 2.3. Lactoferrin and Intestinal Microbiota

As reported by many research groups, LF appreciable effectiveness in preventing or blocking growth of a vast spectrum of pathogens (e.g., *Staphylococcus aureus*, *Listeria monocytogenes*, *Salmonella enterica*, *Streptococcus*, *Legionella pneumophila*, and *Staphylococcus aureus* and *Escherichia coli)* without affecting beneficial bacteria (e.g., *Lactobacillus* and *Bifidobacterium*) has been largely described [61,62]. It would even seem that LF could stimulate the proliferation of probiotic bacteria [63]. The exploration of the LF peptide mechanisms of action for the antibacterial effects mainly revealed (i) the sequestration of bacteria and their deprivation in iron (essential for their survival) by LF given its iron-binding ability [64]; (ii) the potential of LF in interacting with LPS of gram-negative bacteria, thus considerably hampering their growth [8,65,66]; (iii) the LF competency of binding with receptors on microorganisms to obstruct the transport of nutrients, thereby restraining bacterial synthesis and metabolism [67,68]; (iv) the inhibition of microbial enzyme activity by LF [67]; (v) the LF-mediated prevention of biofilm formation by bacteria [69]; (vi) the capacity of LF to release antibacterial peptide lactoferricin and other bioactive peptides following hydrolysis by pepsin [70,71,72]; and (vii) the direct antimicrobial activities exerted by LF [73]. It should be noted that the impact of LF is not limited to bacteria, but can also extend to antivirals [74,75,76], antiprotozoal [77], and antifungal [78] activities. Another important element to remember is that additional LF functions can only be performed in synergy with other components [79,80].

Obviously, the multifunctional LF has the potential to shape intestinal microbiota, a common process in preterm infants, newborns, and children [81]. Consequently, prophylactic administration of LF was observed to prevent late-onset sepsis, necrotizing enterocolitis, and death in preterm infants compared with a placebo [82,83]. Moreover, treatment of mice with IBD prevented the rise in the proportions of *proteobacteria*, *Actinobacteria*, *Cyanobacteria*, *Deferribacteres*, and reversed the proportion of *Firmicutes* compared with those in the control group [59]. The amelioration of IBD-induced mouse microbiota dysbiosis in response to LF treatment comprised reduced *Escherichia* and *Shigella*, while stimulating *Lactobacillus* [59].

### 2.4. Clinical Evidence for Lactoferrin Effectiveness

A significant number of clinical investigations have been carried out particularly in preterm infants to assess the LF impact on intrauterine growth restriction (IUGR). A systematic review and meta-analysis (involving 6 separate studies and 333 pregnancies) reported a diminished risk of preterm birth in women at risk in response to a prophylactic LF intake [84]. Nevertheless, the investigators of this study recommended the need for additional randomized trials of greater size to satisfactorily confirm the contribution of LF to improve perinatal maternal outcomes, particularly the decrease in the risk of preterm birth. Similar to infection, necrotizing enterocolitis and adverse neurodevelopment frequently lead to IUGR. Important investigations have then examined the role of LF on each of these entities, and their data were collectively analyzed by a recent meta-analysis involving 12 clinical trials totaling 5425 preterm infants [85]. The findings emphasized that enteral LF supplementation with or without probiotics lowered sepsis, but only the combination of LF with probiotics showed the potential to decrease GI injury. However, caution should be exercised until the results are confirmed through higher-powered clinical trials taking into account specific parameters, such as concentration, treatment duration, and the source of LF. However, it is very interesting to note the absence of adverse effects of LF administration in preterm babies [86] and the reduction of hospital length of stay [87]. LF may as well improve maternal health (in the perinatal period) by serving as an alternative to antibiotics in treatment of vaginal infection [88], and may at the same time decrease the inflammatory process and ensuing risk of preterm delivery [89]. It should also be underscored that various groups have examined the safety and effectiveness of LF supplementation in preventing/treating other types of infections [90], dermatological conditions [91], *Helicobacter pylori* infection [92], etc. However, a few groups have reported the beneficial effect of oral LF supplementation on microbiota dysbiosis or species richness [25,60,93,94,95,96].

Although some information supporting a beneficial influence on the components of MetS is available, there has been no exhaustive work on CMD in humans as far as we know. An inverse relationship was apparent between circulating levels of LF, and fasting triglyceride and glucose concentrations, waist-to-hip ratio, and in direct correlation with plasma HDL-cholesterol [97]. Similarly, obese women displayed an inverse association between LF and adiposity despite independent positive association with insulin resistance (IR), for which the precise mechanisms for these observations have not been elucidated [13]. Furthermore, LF levels were positively and negatively associated with insulin sensitivity and inflammatory parameters, respectively [19], an important observation since systemic low-grade inflammation commonly characterized individuals with obesity [98,99]. In line with these encouraging LF effects, human LF exhibited the ability to chelate LPS, and to interfere with the binding between LPS and soluble human CD14, thereby alleviating inflammation [100]. Interestingly, the hypolipidemic effect of LF was reported to be dependent on its binding to the LDL receptor-related protein 1 (LRP1). The investigators hypothesized that LF molecule alterations disturb the interaction with LRP1 receptors, which produces repercussions on lipid concentrations and their elimination from blood circulation. In this context, it should be noted that the Arg-rich sequence of LF N-terminus bears resemblance to apolipoprotein E structure sighted by LRP1 [101].

With regard to hypertension, a major component of the MetS, the treatment of peripheral blood monocytes (from normotensive and treated hypertensive individuals) provided the evidence that LF may play a role in hypertension [102]. Even if no human studies are available, basic and preclinical research pointed out the ability of LF-derived peptides to modulate the renin–angiotensin, endothelin-1, and endothelial NO-dependent systems, leading to hypotensive effects [32,33,34,103,104].

Finally, LF and the variants of LF receptor-related genes have been found to be related to metabolic settings in normal and pathological conditions. For example, an investigation conducted in 1749 French Canadians reported a significant difference in allele frequencies between subjects with and without MetS for the *LTF* rs2239692 polymorphism [105]. Another group of researchers documented the significant correlation between the LF rs1126477 gene variant and anthropometric parameters, allowing to propose that subjects with the CT variant of the *LTF* rs1126477 are endowed with a lower waist circumference compared to those with the TT variant [18]. On the other hand, male carriers of the G allele of the LF rs1126478 variant showed significantly elevated HDL-cholesterol and decreased fasting triglyceride concentrations [97]. As to blood pressure, there was a correlation between LF rs1126478 variant and hypertension [102], while pointing out the relation of some SNPs in the sLF gene with the prevalence of metabolic abnormalities and CVD [97,106]. Despite the fascinating broad spectrum of properties and functions of LF, a molecule considered “miraculous” by some researchers, further in-depth research is required to demonstrate its benefits in large-scale clinical studies with longer follow-up.

## 3. General Osteopontin Properties

OPN, amply reported in human and bovine milk, is also detected in body fluids and in small amounts in a broad range of tissues in small amounts. Although OPN is encoded by a single copy gene, it is expressed in varied isoforms with different molecular weights as a result of not only alternative splicing, but also of post-translational modifications, such as glycosylation, serine/threonine phosphorylation, oxidation, tyrosine sulfation, calcium binding, and proteolytic processing, depending on the tissue [107,108]. The OPN glycoprotein, rich in aspartate and sialic-acid residues, displays a spectrum of molecular weights going from 25 to 75 kDa [109,110,111]. This molecular heterogeneity may explain OPN multifunctionality and interactions with several proteins (e.g., integrins and CD44) [112,113,114]. OPN is involved in tissue remodeling, bone morphogenesis, biomineralization, calcification, immune regulation, inflammation, cell signaling, and cytoskeleton rearrangement [115,116,117,118,119]. Diminished pro-inflammatory factors have been observed in infants fed formula supplemented with OPN in comparison with infants fed regular formula [120,121], and a few groups have also reported antioxidant activity [122,123,124] (Figure 2).

### 3.1. Osteopontin and Intestinal Barrier

Human and bovine milk OPN have been reported to be resistant to proteolytic degradation or only partially digested in the gut [151]. Accordingly, integral OPN molecule or its proteolytic products with biological effects have been identified in plasma [152,153]. Former studies underlined the advantageous influence of OPN on the preservation of the gut barrier [154] and epithelial cell survival in view of its anti-apoptotic action [155]. Work on animal models was able to demonstrate OPN capacity to stimulate duodenal mucosal growth, villus thickness, crypt depth, and nutrient absorption [156]. Even more, OPN may largely accumulate on the epithelium surface [157] and exert an attenuation of epithelial injury by blunting NO output in response to macrophage activation [155,158]. Moreover, in vitro and in vivo experiments underscored the “guardianship” of a tight-junction complex by OPN [159]. Likewise, treatment of ethanol-fed mice with milk OPN alleviated disturbances of tight-junction proteins, indicating its potential to provide mucosal protection [159]. It is important to note, however, that very little work has been conducted to provide clear evidence of the role of OPN in gut homeostasis during childhood and adult periods.

### 3.2. Osteopontin and Intestinal Permeability

As ethanol intake facilitates the passage of Gram-negative bacteria from the gut to the portal blood, LPS rises in the circulation, causes inflammatory events, and provokes liver injury [160,161,162]. Based on the demonstrated gut-protective action of OPN, Ge et al. [163] examined whether OPN could maintain intestinal integrity and, consequently, prevent liver inflammation and steatosis. Their findings demonstrated that OPN, in mice with alcohol-induced liver injury, was able to protect gut integrity, while depressing liver pathogenesis. In fact, OPN enhances intestinal mucin production and blunts ethanol-mediated permeability augmentation by maintaining tight-junction integrity [163]. In another intestinal model, OPN prevented injury to the plasma membrane via the increased expression of tight-junction proteins, which decreased bacterial translocation and LPS deposition in the liver [163]. Conversely, OPN deletion in *Opn*^−/−^ mice deteriorated intestinal tissue and repair [115,164], whereas OPN intake mitigated gut damage [153]. Engineered OPN-containing polymeric nanoparticles raised protection of the intestinal mucosal barrier and prevented permeability, while lessening colitis in animal models of IBD [132]. However, it is important to note the lack of studies on intestinal permeability in the adult period and, particularly, in the cardiometabolic field.

### 3.3. Osteopontin and Intestinal Microbiota

A considerable study has explored the multifaceted involvement of OPN in multiple biological processes, bone remodeling, and inflammation [165]. However, less is known about OPN interaction with microbiota.

As mentioned above, OPN-deficient mice showed the importance of OPN in potent development of T-helper 1 immune responses, suggesting its critical function in fighting microbial and viral infection [166,167]. Although not all studies are unanimous on the role of OPN in the control of intestinal inflammation, OPN knockout fast-tracked the development of spontaneous mouse colitis in response to gut microbiota dysbiosis [134]. Differences were noted in enteric bacterial composition, including the augmented abundance of the *Clostridium* cluster and the diminished abundance of the *Clostridium* subcluster XIVa, which earlier displayed a protective impact on murine models [168]. Previous investigations revealed the high effectiveness of human macrophages to phagocytose *E*. *coli* under OPN influence [169]. In fact, OPN favors macrophage phagocytosis, culminating in destroying bacteria and fungi [135]. Overall, OPN strongly bound to various bacteria via its actions as an opsonin, which stimulated macrophage phagocytosis [170]. The mechanism for the preservation of gut barrier and liver homeostasis in animal-administered alcohol could account for the beneficial modification of intestinal microbiota, thereby leading to the production of tryptophan metabolites and short-chain fatty acids. Thus, OPN acts as a multifunctional cytokine that can exert both pro- and anti-inflammatory effects, depending on physiological and pathophysiological conditions. This duality may suggest divergent intestinal microbiota regulation.

### 3.4. Clinical Evidence for Osteopontin Effectiveness

As mentioned previously, clinical trials provided the evidence that supplementation of infant formulas with bovine OPN was safe and well absorbed [171], lowered inflammatory cytokines, and displayed potency in decreasing the prevalence of fever [121,172]. In a preclinical investigation using the pig model, OPN administration led to slight ameliorations in gut structure and systemic immunity [173]. Furthermore, OPN supplementation could prevent necrotizing enterocolitis in preterm piglets [174].

The influence of weight loss was assessed on OPN status in children and adolescents, who were on a diet and exercise program [175]. The results revealed a positive relationship between decreased endogenous OPN and diminished body mass index, which confirms the data of a study documenting OPN elevation in overweight and obese subjects [176]. Interestingly, higher OPN levels were detected in the adipose tissue of morbidly obese patients in association with macrophage infiltration [177,178]. Likewise, mouse obesity resulting from a high-fat diet regimen was characterized by high plasma OPN concentrations and enhanced OPN expression in macrophages recruited into adipose tissue [117]. On the other hand, other reports pointed out the lack of correlation between plasma OPN and waist circumference or body mass index [179,180,181]. This is in line with adipocyte cultures, which exhibited modest effects after their exposure to OPN [182]. Similarly, the induction of OPN and its receptor CD44 in the liver was locally related to IR and liver damage [178]. It is possible that OPN overexpression in hepatic natural killer cells led to endoplasmic reticulum stress and JNK hyperactivation, thereby impairing insulin signaling in the liver [143]. Various groups also highlighted the association of OPN with diabetes, metabolic associated fatty liver disease, stroke, and carotid atherosclerosis [183,184,185,186,187].

OPN may act as an immune system enhancer, promoting infant and child development. Furthermore, OPN exhibits the capacity to protect the intestinal mucosal barrier while maintaining the ecological balance of gut microbiota, thereby preserving cardiometabolic health. Nevertheless, if OPN has been instrumental for various physiological processes, it has been associated with a broad range of pathophysiological conditions, especially in the adult stage. OPN was considered as a central mediator of atherosclerotic plaque development and arterial calcification [188,189,190,191]. In addition, in subjects with coronary artery disease and peripheral artery disease, elevated OPN concentrations could predict future cardiovascular death and long-term adverse outcomes, respectively [192,193]. However, not only were other groups unable to detect any association between circulating OPN and coronary artery disease degree or severity [194], but inconsistencies and contradictions were reported. In addition, beneficial actions of OPN have been highlighted, including cytoprotective impact on cardiac endothelial cells along with their survival in stress conditions [195,196], and inhibition of their apoptosis [195,197]. In keeping with favorable OPN findings, OPN-knockout mice revealed defective myocardial angiogenic response post-myocardial infarction, thereby culminating in detrimental left ventricle remodeling [198], which clearly indicates OPN function in restoring myocardial capillarization in infarcted myocardium. In the same line, OPN has recently been found to promote infarct repair via the amelioration of scar formation and cardiac function [138]; and, on the other hand, OPN knockout mice exhibited vulnerability in developing post-myocardial infarction left-ventricular-chamber dilatation [199].

Indeed, tissue infiltration of macrophages as observed in obesity is dependent on the expression of OPN, which promotes monocyte chemotaxis and motility. Recently, Nomiyama et al. [117] demonstrated that mice after a high-fat diet exhibited increased circulating OPN levels. Obese mice lacking OPN showed improved insulin sensitivity and decreased macrophage infiltration into adipose tissue. These experiments add OPN to a long list of pro-inflammatory pathways involved in the development of IR.

The same discords apply to the role of OPN in CMD. If OPN was associated in the development of adipose tissue inflammation and IR [117,200], obesity, and hepatic steatosis [201,202], there was an unaltered fatty acid oxidation and synthesis between wild-type and *OPN* knockout livers in obesity (along with the downregulation of Forkhead box O1 and PPAR gamma co-activator 1α), which usually enhance mitochondrion genesis, lipid degradation, and insulin sensitivity [142]. Other reports underline that *OPN* deficiency in old mice led to hepatic steatosis and hypertriglyceridemia in relationship with IR [136]. Similarly, liver fibrosis was noted in *OPN*-knockout mice under high-fat feeding in association with inflammation, DNA damage, and IR [136]. The hepatic protective role of OPN was emphasized in alcoholic hepatitis [203,204] and in transgenic mice, which markedly lowered hepatic steatosis, balloon cell degeneration, lipid peroxidation, inflammation, and plasma alanine aminotransferase [133].

### 3.5. Attempts to Explain the Areas of Uncertainty and Divergence Observed in the Numerous Studies

The first observation that can be made is that surprisingly levels of OPN have not been established in healthy people, and, furthermore, the cut-off values are quite arbitrary in most scientific reports, which may have an impact on the interpretation of the findings. What further complicates the picture is that investigators measured total OPN expression, without taking into account the three OPN isoforms resulting from human OPN splicing (OPN-a, OPN-b, and OPN-c) [205]. To this should be added divergences in experimental models and the methodology for determining OPN, thereby explicating the huge heterogeneity of the outcomes. Probably also, the limited number of specimens/animals, the small size of the cohorts, and the fluctuating follow-up duration must have significantly favored the great variability of the results. Other important drawbacks include the cause-effect relationship between OPN and clinical outcomes, as well as the interference of different pharmaceutical agents that the patients were already on at the time the treatment when OPN was added. Finally, the discordant effects of OPN may be attributed to its large groups of variants that happen as a consequence of transcriptional, posttranscriptional, and post translational modifications; and include phosphorylation, proteolytic cleavage, sialylation, and transglutaminase cross-linking. These OPN variants may be specific to the type of organs, tissues, and cells. It is also worth noting that CD44, the receptor of OPN, also presents various isoforms that probably are tissue-specific in accordance with their OPN ligands of a polymorphic nature [206]. Clearly, studies to date have not taken into account the specificities of intra-tissue or cellular OPN and CD44. Definitely, additional work is needed to elucidate these exciting aspects.

## 4. General Lactoferrin–Osteopontin Properties

The multifunctional LF and OPN molecules have a high affinity for each other due to their opposite charge [207], and form a complex in native human and bovine milk [208]. Even in vitro, purified LF and OPN proteins can form an LF–OPN complex with high affinity, driven by electrostatic forces [209]. The LF–OPN complex binds to LF and/or OPN receptors on the cell membrane of intestinal epithelial cells, thus exerting similar or stronger effects on the intestine compared to individual proteins of this complex [210]. In fact, the LF–OPN complex exhibited a higher ability to stimulate proliferation and differentiation of intestinal cells than LF and OPN separately [210]. The LF–OPN complex is resistant to GI digestion, crosses the intestinal barrier without degradation, and reaches in intact from the bloodstream [210,211]. In addition, the LF–OPN complex triggered an enhanced secretion of the IL-18 intermediate between LF and OPN [210], which protects the intestinal epithelium from inflammation [212] (Table 1).

### 4.1. LF–OPN Complex, Intestinal Barrier, and Microbiota

Despite the intense efforts that have been deployed on each of the LF and OPN proteins in relation to the intestinal barrier and gut microbiota, there is virtually no work on the influence of the LF–OPN complex. Astonishingly, limited evidences are reported by the scientific literature concerning the impact of the complex. In this context, the exposure of Caco-2 cells to the LF–OPN complex resulted in a marked secretory stimulation of IL-18 [210]. This cytokine is known to produce a protective effect against enterocyte inflammation and colitis [213,214], which is supported by the susceptibility of *IL-18 knockout* mice and *IL-18 receptor knockout* mice to colitis [214]. Both IL-18 and IL-18r seem necessary for intestinal barrier integrity, epithelium homeostasis, mucosal repair mechanisms, and appropriate host interaction with the gut microbiota [213,215]. However, it is essential to obtain direct proof of concept by studying the effect of the LF–OPN complex on barrier function, especially in view of the contradictions reported by other studies [216,217]. There is also a need to delineate the mechanisms of the antibacterial activity and function (as noted for enteropathogenic *Escherichia coli*) in response to the LF–OPN complex [210].

### 4.2. Clinical Evidence for LF–OPN Complex Effectiveness in Cardiometabolic Diseases

In spite of their praiseworthy features of LF and OPN individually, to our knowledge, no study has yet been reported examining their combinative effects on CMD. Therefore, additional investigations are required to tackle the following issues: Does the LF–OPN complex alleviate OxS and inflammation, which contribute to IR, a factor promoting the clinical manifestations of MetS components? Would the LF–OPN complex be capable of blunting obesity, dyslipidemia, metabolic endotoxemia, and gut–liver axis pathogenesis (responsible for the development and progression of metabolic associated fatty liver disease)? Is the metabolic health improvement mediated by intestinal barrier function and a gut microbiota-dependent mechanism?

## 5. Conclusions

Intense research is underway to develop new therapeutic strategies in the treatment and management of CMD. Attention is specially paid to new therapeutic strategies that target their causal factors. Currently, a growing attention is devoted to human breast and bovine milk proteins, which are regarded as indispensable health-promoting foods. This review has outlined the current knowledge of two multifunctional bioactive milk proteins, LF and OPN, which can bind together to form a powerful LF–OPN complex with more beneficial effects. Considerable emphasis has initially been placed on the physiological, biochemical, and nutritional functionalities of these biologically active compounds, while revealing their advantageous effects on the development of preterm babies, neonates, and infants. Antioxidant, anti-inflammatory, immunomodulatory, antibacterial, antifungal, and antiviral activities explain the highly protective and growth-promoting mechanisms of these milk-derived proteins. In a second step, an updated overview was provided on how the latter shape intestinal homeostasis through the reinforcement of intestinal mucosal barrier and gut microbiota, two robust guardians of the cardiometabolic health. Finally, a critical analysis of preclinical and clinical studies was presented documenting the influence of each bioactive protein on CMD. Particular attention should be devoted to the administered dose of LF and OPN. By examining the literature, one can quickly notice that the doses of lactoferrin or osteopontin are very variable in the different preclinical and clinical trials. Some groups administered them orally, intraperitoneally, or systemically. Others delivered them in a single dose or gave them over several days with or without consideration of body weight. These differences make comparisons difficult and explain the discrepancies in results. In addition, the relationship between the dose administered and blood concentration is very poorly documented in clinical trials. This is a set of important points that require the attention of the scientific community.

## Figures and Tables

**Figure 1 nutrients-15-01394-f001:**
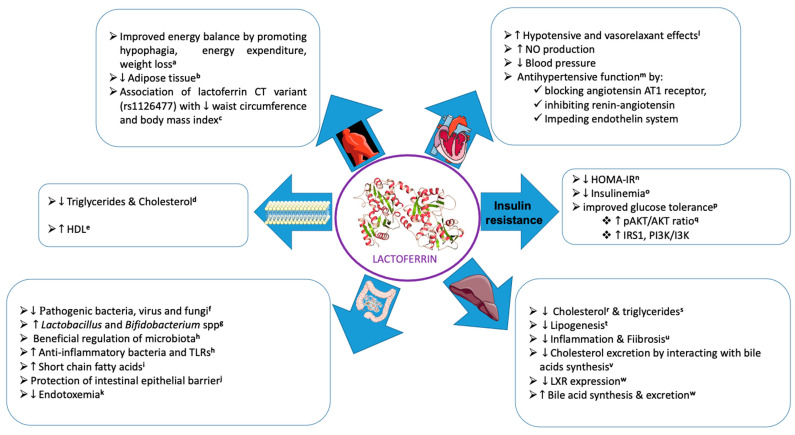
Pleiotropic functions of bioactive milk lactoferrin through several mechanisms. (The images used were extracted from Servier Medical Art). Related references cited in the Figure: (**a**: [11,12,13,14,15,16]; **b**: [16,17]; **c**: [18]; **d**: [11,19]; **e**: [19]; **f**: [20,21]; **g**: [20,22]; **h**: [23]; **i**: [24,25,26]; **j**: [25,27,28,29]; **k**: [30,31]; **l**: [32]; **m**: [33,34]; **n**: [11,19]; **o**: [15]; **p**: [11,15,19]; **q**: [16,35,36]; **r**: [16,37]; **s**: [11,35,38]; **t**: [16,38]; **u**: [16]; **v**: [39]; **w**: [40]).

**Figure 2 nutrients-15-01394-f002:**
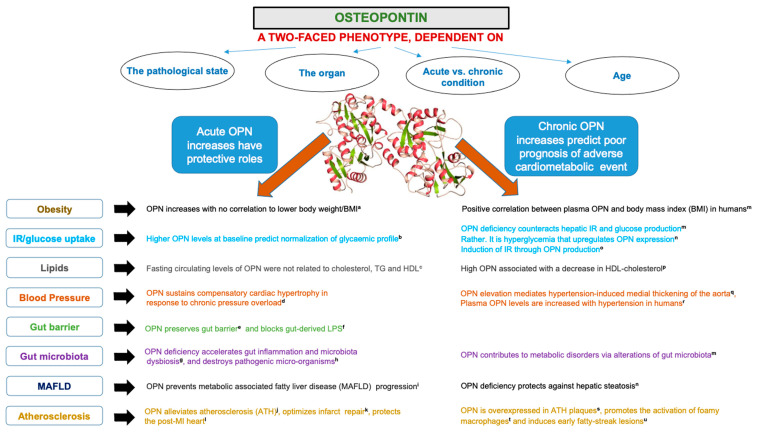
Biological functions of bioactive milk osteopontin. Related references cited in the Figure: (**a:** [125,126]; **b**: [127]; **c**: [128]; **d**: [129]; **e**: [130,131,132]; **f**: [133]; **g**: [134]; **h**: [135]; **i**: [136]; **j**: [137]; **k**: [138]; **l**: [139,140]; **m**: [141]; **n**: [142]; **o**: [143]; **p**: [144]; **q**: [145]; **r**: [146]; **s**: [147]; **t**: [148,149]; **u**: [150]).

**Table 1 nutrients-15-01394-t001:** Characteristics of lactoferrin, osteopontin, and lactoferrin–osteopontin complex.

	Lactoferrin (LF)	Osteopontin (OPN)	LF + OPN
**Molecule**	Glycoprotein	Glyco-phosphoprotein	
**MW**	80 kDa	33.9 kDa	
**Amino acids**	703	298	
**Source**	Milk, mucosal, secretions, neutrophils	Milk, diverse cell types, tissues, organs	Milk
**Polymeric form**	✔✔✔	✔	
**GI digestion**	Resistant	Resistant	Resistant
**Electric charge**	Negative	Positive	
**Binding to**	Iron	calcium	
**Antipathogenic**	Bacteria, virus, fungi	Bacteria, virus, fungi	Bacteria, virus, fungi
**Protection of gut** **barrier**	✔	✔	✔
**LPS sequestration**	✔	✔	✔
**Oxidative stress**	Antioxidant	Antioxidant	Antioxidant
**Inflammation**	Anti-inflammatory	Anti-inflammatory	Anti-inflammatory
**Immuno-protection**	✔	✔	✔
**Intestinal epithelialproliferation**	✔	✔	✔✔
**Proliferative** **Mechanisms**	MAPK	PI3K/Akt	PI3K/Akt
**Intestinal epithelial** **cell differentiation**	✔	✔	✔✔
**Receptors**	CD14, LRP-1, Intelectin-1, TLR4, CXCR4, HSPGs	Integrin, CD44	

Akt: Protein kinase B; CXCR4: C-X-C chemokine receptor type 4; HSPG: Heparan sulfate proteoglycan; LRP-1: Low density lipoprotein receptor-related protein-1; MAPK: Mitogen-activated protein kinase; PI3K: Phosphoinositide 3-kinase; TLR4: Toll-like receptor 4.

## Data Availability

Not applicable.

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
