# Peer review of "Lactoferrin, Osteopontin and Lactoferrin–Osteopontin Complex: A Critical Look on Their Role in Perinatal Period and Cardiometabolic Disorders"

_nutrients, 2023, doi:10.3390/nu15061394_

Round 1

Reviewer 1 Report

Comments

1.    The authors have done a good job of describing the role of general properties and functions of each of the specific milk-derived bioactive components.

2.    The authors here delve into the role of human and bovine milk but how about soy, almond, and oat milk which are suddenly gainly popularity? Do these milk types have the same general properties of human and bovine milk? Can the authors do a comparison here?

3.    A better arrangement of the text is definitely required and I have highlighted that below. Please also explain the clinical studies in the form of a table rather than long paragraphs. It becomes easier for the reader to follow that way.

4.    I would also segregate mouse and human studies that way it is not a mixed bowl of information for the reader.

5.    Why is the effect of osteopontin on intestinal barrier and intestinal permeability separate topics. That need to be combined. Also the authors should come up with a better subtitle for this section.

6.    You can’t have a question as a subsection. Please rephrase. Section3.5

7.    Please re-draw Figure 1. There is a barrage of information in a figure which is not required. Trim the fat and make the figure presentable. Also, what are the references below each figure? It needs to be more presentable.

Author Response

  1. The authors have done a good job of describing the role of general properties and functions of each of the specific milk-derived bioactive components.

We thank the Reviewer for the appreciation of our work.

  1. The authors here delve into the role of human and bovine milk but how about soy, almond, and oat milk which are suddenly gainly popularity? Do these milk types have the same general properties of human and bovine milk? Can the authors do a comparison here?

As well specified in the manuscript, the central objective of the present paper is to specifically review the role of lactoferrin and osteopontin, two major human and bovine milk proteins, gaining popularity in the scientific community. In particular, these molecules independently or conjointly influence the development of cardiometabolic disorders, which constitutes the preponderant reason we chose them. Our goal was not to compare the properties of different milks, but to focus mainly on these two proteins, specifically found in high concentrations in human and bovine milk.

  1. A better arrangement of the text is definitely required and I have highlighted that below. Please also explain the clinical studies in the form of a table rather than long paragraphs. It becomes easier for the reader to follow that way.

Given the limited number of clinical studies and the variety of their content, it is difficult to organize the clinical studies in tabular form. For this reason, we have opted for the succinct writing of outcomes. 

  1. I would also segregate mouse and human studies that way it is not a mixed bowl of information for the reader.

The clinical studies section as a whole refers to work on humans, not animals.

  1. Why is the effect of osteopontin on intestinal barrier and intestinal permeability separate topics. That need to be combined. Also the authors should come up with a better subtitle for this section.

We believe that separating the two sections is advantageous to readers given the unique characteristics of each. The section on the intestinal barrier refers primarily to physicochemical properties, whereas the section on permeability deals with the function of the barrier. We therefore prefer to keep the two parts separate unless the editor decides otherwise.

  1. You can’t have a question as a subsection. Please rephrase. Section3.5

We have eliminated the question as a title in favor of a classic title.

  1. Please re-draw Figure 1. There is a barrage of information in a figure which is not required. Trim the fat and make the figure presentable. Also, what are the references below each figure? It needs to be more presentable.

We believe that Figure 1 represents a good illustration for the reader since it documents the effects of lactoferrin on various organs and under different conditions. In fact, we have worked hard to obtain a figure with concentrated information for the reader, which allows visualizing at once the action of lactoferrin along with the pertinent references (within the reach of the readers), the impact on different organs, and the mechanisms of action. However, as the Reviewer suggested, we have removed the boldface and improved the references.

Reviewer 2 Report

This review paper by Levy and Coll. is aimed at discussing the role of LF and OPN and potentially of their complex in 2 conditions of human life/health: perinatal period and cardiometabolic diseases.

The topic is interesting and the two areas covered by the paper are both important and relevant to human health.

The text is clearly written and understandable. I suggest a minimal check of the English language (see also below some minor comments).

I suggest to integrate the paper as follows.

1. Introduction: it should briefly also mention the concept of LF-OPN complex and its potential significance, as it will be discussed in the last part of the paper.

2. General lactoferrin properties: LF is discussed in the paper as both of human origin (taken up by human milk during lactation and then apparently produced in the body, as it can be measured in the circulation) as well as of other origin (bovine, recombinant-human??) and used in the form of food supplement or even released by engineered probiotics, again taken as food supplement.

All these details, including the differences between human and bovine LF should be clearly discussed. let's not forget that the journal Nutrients is interested in reporting details on these type of compunds useful as food supplemnts/nutraceuticals. Please also report information on actual levels and measurement of LF in the circulation (across ages, sexes etc.). I would separate the part regarding LF use as supplement from that of LF levels as biomarker

3. General OPN properties: also here, dicuss in greater detail endogenous levels and possibly exogenous OPN administration.

4. this section is clearly much shorter due to the substantial lack of some information/studies. It is anyway a novel and relevant piece of information pertinent to the paer.

Specific comments

LINE 35 change to:  ….as the expression of many more than 10000 genes were

LINE 171 correct as:  …has been..  

LINE 419 evidence is uncountable …correct to evidence…

Author Response

This review paper by Levy and Coll. is aimed at discussing the role of LF and OPN and potentially of their complex in 2 conditions of human life/health: perinatal period and cardiometabolic diseases. The topic is interesting and the two areas covered by the paper are both important and relevant to human health. The text is clearly written and understandable. I suggest a minimal check of the English language (see also below some minor comments).

We thank the Reviewer for his encouraging words

  1. Introduction: it should briefly also mention the concept of LF-OPN complex and its potential significance, as it will be discussed in the last part of the paper.

 We followed the Reviewer's recommendations and mentioned the Lf-OPN concept in the Introduction.

  1. General lactoferrin properties: LF is discussed in the paper as both of human origin (taken up by human milk during lactation and then apparently produced in the body, as it can be measured in the circulation) as well as of other origin (bovine, recombinant-human??) and used in the form of food supplement or even released by engineered probiotics, again taken as food supplement. All these details, including the differences between human and bovine LF should be clearly discussed. Let's not forget that the journal Nutrients is interested in reporting details on these type of compounds useful as food supplemnts/nutraceuticals. Please also report information on actual levels and measurement of LF in the circulation (across ages, sexes etc.). I would separate the part regarding LF use as supplement from that of LF levels as biomarker.

The points suggested by the Reviewer seem important. Therefore, we have elaborated on them as recommended.

  1. General OPN properties: also here, discuss in greater detail endogenous levels and possibly exogenous OPN administration.

This is a very important question, but unfortunately it cannot be answered correctly because of the discrepancies between studies, whether preclinical or clinical. We have highlighted this important point in the conclusion : ‘’Particular attention should be devoted to the administered dose of LF and and OPN. By examining the literature, one can quickly notice that the doses of lactoferrin or osteopontin are very variable in the different preclinical and clinical trials. Some groups administered them orally, intraperitoneally or systemically. Others delivered them in a single dose or gave them over several days with or without consideration of body weight. These differences make comparisons difficult and explain the discrepancies in results. In addition, the relationship between the dose administered and blood concentration is very poorly documented in clinical trials. This is a set of important points that require the attention of the scientific community’’.

  1. This section is clearly much shorter due to the substantial lack of some information/studies. It is anyway a novel and relevant piece of information pertinent to the paper.

 This section is short because the information available in the literature is limited.

Specific comments

 LINE 35 change to:  ….as the expression of many more than 10000 genes were

 The correction has been made

LINE 171 correct as:  …has been..  

 The correction has been made

LINE 419 evidence is uncountable …correct to evidence…

The correction has been made

Reviewer 3 Report

Very good review on lactoferrin, osteopontin and their bioactive complex. The Authors are focused on functional properties of these proteins, looking at physiological, biochemical and nutritional properties. Their special interest is how these proteins influence health in perinatal period. Particularly important is their ability to control inflamation and oxidative stress. The Authors pay attention on gut mucosal barrier and intestinal microbiota, especially as the important factors linked to cardiometabolic disorders. Health issues connected with obesity, insulin resistance, dyslipidemia, blood hypertension, diabetes and atherosclerosis are discussed. Lactoferrin, osteopontin and their bioactive complex can have very important impact on prevention and therapy of these cardiometabolic disorders. The text is very well written. English level is excellent. 176 positions of literature are quoted, what makes this review one of the most comprehensive review on physiological, biochemical and nutritional properties of lactoferrin, osteopontin and their bioactive complex.

Author Response

Our gratitude to the Reviewer for his encouragement and praise 

Round 2

Reviewer 2 Report

All comments have been properly addressed. At line 404 "evidences" should be written "evidence" (uncountable).